# CuES: Bottom-Up Exploration and Top-Down Guidance for Agentic Data Synthesis

## Abstract

Training LLM-based agents with reinforcement learning (RL) in complex environments requires high-quality, environment-specific data. However, generating tasks that are semantically coherent, behaviorally valid, and executable is prohibitively expensive, making the scarcity of such data a fundamental bottleneck for scaling capable agents. Existing synthesis methods struggle to balance high-level intent with environmental grounding, often producing either unexecutable instructions or aimless, low-quality trajectories. To address this dilemma, we propose **CuES**, a **Cu**riosity-driven and **E**nvironment-grounded framework for agentic data **S**ynthesis that operates without predefined queries. CuES first uses curiosity-driven exploration to uncover a foundation of fundamentally solvable interaction patterns, ensuring executability by design. Concurrently, top-down guidance expand exploration and task diversity while keeping generated tasks aligned with user intentions. Experiments on AppWorld, BFCL, and WebShop show that CuES generates diverse, executable, and high-quality training tasks, achieving or surpassing the diversity and effectiveness of manually curated datasets and delivering strong downstream RL performance, which makes it possible to train environment-specific agents cost-effectively and efficiently. The code is available at `https://github.com/Anonymize-Author/CuES`.

## 1 Introduction

As LLM-based agents are increasingly deployed in complex interactive environments—from graphical user interfaces to open-ended web platforms—obtaining high-quality, environment-specific training data becomes crucial. This data must precisely capture the target environment's dynamics and constraints to support effective agent training, particularly through reinforcement learning (RL) which optimizes behavior via environment feedback (Shang et al., 2025; Li et al., 2025b). The quality of the underlying task or query data is paramount: effective queries must demonstrate **semantic coherence** through clearly defined objectives and intents; exhibit **behavioral validity** by respecting the environment's rules and affordances; and ensure **executability** with actions the agent can realistically perform (Mai et al., 2025; Singh et al., 2025). However, manually crafting queries meeting all these stringent criteria demands significant domain expertise and annotation effort, rendering large-scale collection prohibitively costly. Consequently, the scarcity of diverse, high-quality, environment-specific task data constitutes a fundamental bottleneck to scaling capable RL agents for real-world applications (Cheng et al., 2024; Zhang et al., 2024).

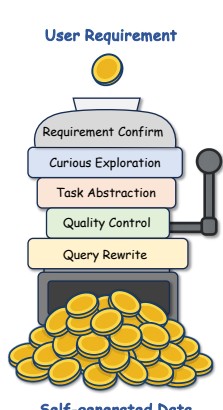

Figure 1: Structure of CuES. Just provide user requirement and a large amount of data can be synthesized like a slot machine.

To address this data scarcity, existing work generally follows two complementary lines. **Top-down approaches** start from human-written seed queries or LLM proposals and expand them into task instructions. This strategy benefits from clear goal specification and controllable semantics, but

often *decouples task generation from environment dynamics*, producing instructions that look plausible in text yet fail during execution(Li et al., 2024; Lù et al., 2024; Lai et al., 2024). In contrast, **bottom-up approaches** explore the environment first and then abstract tasks from discovered trajectories, ensuring that proposed problems are feasible within the environment. While this improves executability and behavioral grounding, it typically suffers from weak goal alignment, inefficient exploration, and domain-specific heuristics that limit transferability. Moreover, the current exploration of bottom-up approaches only target a few narrow areas such as web search, and the implementation is also highly coupled to specific tasks, which makes it difficult to be applied in practice(Sun et al., 2025). **The persistent divide between the goal clarity of top-down pipelines and the environmental grounding of bottom-up exploration calls for a framework that strategically fuses both forces, enabling the synthesis of data that are diverse, executable, and precisely aligned with downstream objectives**.

To bridge this divide, we present **CuES**, a **Cu**riosity-driven and **E**nvironment-grounded framework for agentic data **S**ynthesis in query-free or limited query environments. CuES operates without predefined tasks or prompts and unfolds through five coordinated stages—requirement confirmation, curiosity-guided exploration, task abstraction, quality control, and query rewriting—as illustrated in Fig. 1. Its central trajectory follows a bottom-up paradigm in which intrinsic curiosity drives the agent to explore the environment and uncover solvable interaction patterns that are subsequently abstracted into reusable task formulations. Building on this bottom-up exploratory backbone, CuES integrates top-down complementary mechanisms *Environment Memory* and *Concept Pools* that couple bottom-up discovery with top-down control. Through the interplay of intrinsically motivated exploration and top-down control, CuES achieves a synthesis process that is both diverse and executable while remaining closely aligned with the requirements of robust agent training. We evaluate CuES on three agentic benchmarks including AppWorld(Trivedi et al., 2024), BFCL(Patil et al., 2024), and WebShop(Yao et al., 2022). Across these environments, synthesized data achieve diversity and effectiveness on par with or surpassing original datasets, and they translate into strong downstream utility for agent training. **These results suggest that an environment grounded, curiosity driven process, when coupled with lightweight top down guidance and memory, can produce high quality data without relying on predefined queries or external seed queries.**

The main contributions of this work are summarized as follows:

- It introduces *CuES*, a curiosity-driven framework for agentic data synthesis without predefined seed queries or external corpora. Through a five-stage process, it automatically generates high-quality, executable, and diverse training task.

- It incorporates two **top-down** mechanisms into a **bottom-up** backbone. *Environment Memory* records salient states for bottom-up exploration to revisitation and reuse, while *Concept Pools* conditions it on training objectives to guide curiosity and improve efficiency.

- Experiments on AppWorld, BFCL, and WebShop show that CuES produces data with diversity and executability comparable to or exceeding manually curated datasets and delivers strong downstream training performance.

## 2 RELATED WORK

We organize related work into two threads: (i) top-down imitation-based synthesis that mimics existing data, (ii) bottom-up exploration that discovers tasks from interaction. CuES sits at the intersection: it is query-free and bottom-up in how tasks emerge from interaction, while injecting lightweight top-down guidance via Environment Memory and Concept Pools, and enforcing executability with explicit judging.

### 2.1 TOP-DOWN IMITATION-BASED SYNTHESIS: MIMICKING EXISTING DATA

Top-down pipelines typically rely on human-authored tasks or expand a small seed set using LLMs(Wu et al., 2025; Li et al., 2025a). Recent web and GUI agents illustrate both the promise and the pitfalls of this approach. Systems such as AutoWebGLM (Lai et al., 2024) and WebLINX (Lù et al., 2024) operate with explicit high-level goals or dialogue instructions and often imitate patterns present in existing benchmarks. While this improves goal clarity, it tends to decouple proposals from environment dynamics, leading to low executability when intermediate preconditions

are missing or tool affordances are misidentified. Empirical analyses show that scale alone does not resolve these issues: UI control studies find that simply increasing dataset size does not guarantee coherent, solvable trajectories (Li et al., 2024), and self-improvement strategies without grounded verification may propagate subtle errors across steps (Patel et al., 2024).

The core limitations of imitation-based synthesis are twofold. First, dependence on predefined high-level tasks restricts scalability and curtails diversity, since the proposal space is ultimately bounded by seed patterns (Lai et al., 2024). Second, quality is hard to ensure: early-step mistakes or mismatched goals can corrupt entire trajectories, yielding incomplete or incoherent data even when text instructions look plausible (Li et al., 2024; Patel et al., 2024). These issues form a bottleneck for advancing agents from scripted automation to robust autonomy in real environments.

**CuES differs by removing reliance on manual seed queries and decoupling quality from imitation.** Tasks emerge bottom up from witnessed interactions, then pass through explicit execution and judgment before any rewriting. Lightweight top-down signals—requirement confirmation and concept pools—serve to shape coverage and reduce wasted exploration, not to prescribe solutions. The result is a synthesis pipeline that attains the clarity often associated with top-down methods while maintaining the executability and groundedness of bottom-up discovery.

### 2.2 BOTTOM-UP EXPLORATION: DISCOVERING TASKS FROM INTERACTION

A growing line of work synthesizes data directly from environment interaction, reversing the conventional "specify task then collect trajectory" pipeline. OS-Genesis (Sun et al., 2024) exemplifies this reversal for GUI agents: an agent first explores the desktop and records raw traces, then retrospectively derives high-level tasks from the discovered behaviors, filtering trajectories to ensure quality and solvability. By grounding tasks in what the environment can actually support, this paradigm improves executability and increases variety relative to task-first scripting. These systems highlight a central advantage of bottom-up synthesis: proposals are feasible by construction because they originate from witnessed interactions.

Despite these gains, bottom-up pipelines face key challenges. Without sufficient guidance, exploration can drift, producing many trajectories that lack clear purpose or fail midway, which lowers overall yield and inflates redundancy (Murty et al., 2024). Moreover, many implementations are tailored to specific domains (e.g., GUI/web navigation), raising concerns about cross-domain generalization and transferability of the synthesized corpora (Sun et al., 2024). In practice, the absence of simple mechanisms to align discovery with desired coverage or stage-specific training needs can lead to substantial ineffective exploration and narrow gains outside the source domain.

**CuES preserves the executability benefits of bottom-up discovery while addressing these limitations.** It remains query-free and interaction-driven but introduces requirement confirmation and concept pools to softly aim exploration toward salient regions of the environment without prescribing exact tasks. An environment-indexed memory stores compact state–action sketches, enabling the system to revisit informative frontiers and avoid redundant loops. Crucially, only validated successes proceed to rewriting, which broadens diversity while keeping the behaviors executable.

## 3 METHOD

**Problem Formulation** As illustrated in Fig. 2, the goal of CUES is to synthesize executable and training-valuable tasks from an interactive environment. We formalize this as

$$F : (E, M, U, S) \longrightarrow \mathcal{T}_{\text{synt}},$$

where $E = \{t_1, \dots, t_m\}$ is an executable environment including a finite tool set, $M$ is a structured environment description, $U$ is an optional user requirement, and $S$ is an optional seed-query set. The output $\mathcal{T}_{\text{synt}}$ denotes a vetted set of reliable tasks.

A task is a pair $(q, \tau)$, where $q$ is a natural-language query and $\tau = \left( (s_t, a_t, o_t) \right)_{t=1}^{T}$ is a trajectory of states $s_t$, admissible actions $a_t \in \mathcal{A}(s_t)$, and observations $o_t$. Executability means that every action $a_t$ is valid in state $s_t$ and leads to the recorded transition.

The mapping $F$ is decomposed into five stages: requirement confirmation derives guiding principles $\mathcal{P}$ and a concept pool $\mathcal{C}$ from $(E, M, U, S)$; curious exploration interacts with the environment under

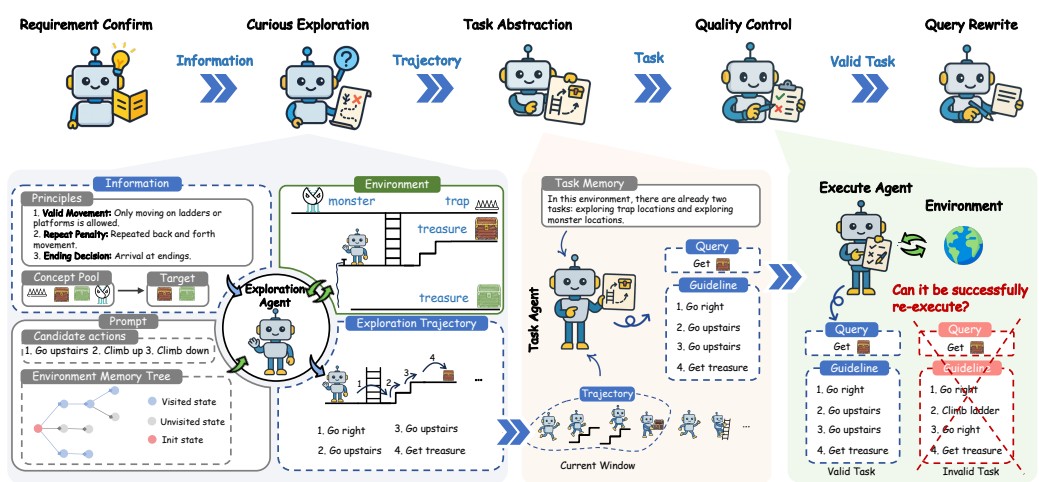

Figure 2: CuES pipeline. (a) Requirement Confirm constructs the concept pool $\mathcal{C}$ and principle $\mathcal{P}$ by extracting concepts from the environment description $M$ and seed queries $S$ and filtering them with the user need $U$. (b) Curious Exploration executes candidate actions conditioned on $(\mathcal{P}, \mathcal{C})$, consults the environment memory tree to prioritize unseen actions, and emits triples $(s, a, o)$ (eq.3.2) as exploration trajectories. (c) Task Abstraction groups consecutive triples within a batch into executable queries with guidelines. (d) Quality Control re-executes each query. (e) Query Rewrite progressively exposes guideline hints in the query text to lower difficulty.

these constraints, with memory $\mathbb{M}$, to collect trajectories $\Pi$; task abstraction consolidates trajectories into candidate tasks $\mathcal{T}_{\text{cand}}$; quality control re-executes and filters candidates to obtain $\mathcal{T}_{\text{synt}}$; and query rewrite reformulates queries to adjust difficulty and enrich variety. Full definitions are given in Appendix D.

## 3.1 REQUIREMENT CONFIRMATION

As shown in Fig. 2(a), the requirement confirmation stage takes three inputs: an environment description $M$, an optional user need $U$, and an optional seed query set $S$. It produces two outputs: a **concept pool** $\mathcal{C}$, which grounds subsequent exploration, and a set of **actionable principles** $\mathcal{P}$, which is extracted by principle agent through environment description and concept pool and specify output schema and highlight priority actions. Both outputs are passed forward to guide the next stage.

**Concept Pool** To initialize exploration, we construct a preliminary concept pool by combining concepts extracted from the environment and from the seed set:

$$\tilde{\mathcal{C}} = U(\underbrace{\Phi_{\text{env}}(M)}_{\text{from environment description}} \cup \underbrace{\Phi_{\text{seed}}(S)}_{\text{noun phrases and action}}). \tag{1}$$

Here, $\Phi\text{env}(M)$ identifies entities and affordances from $M$ (e.g., categories, tools, admissible actions), while $\Phi_{\text{seed}}(S)$ extracts noun phrases and action predicates when $S$ is provided. If $M$ is absent, $\Phi_{\text{env}}(M)$ falls back to a domain-wide inventory; if $S$ is absent, $\Phi_{\text{seed}}(S) = \varnothing$. When a user need $U$ is specified, it filters the pool to obtain the final concept set, $U$ is a agent filter executed according to user need at this time; otherwise, the preliminary pool remains unchanged. Detailed examples refer to Fig.2(a).

Principles $\mathcal{P}$ are carried forward into the next stage's prompt to specify the desired output format and to highlight the main action families to explore, while $\mathcal{C}$ anchors exploration in the entities and actions admissible in the environment.

## 3.2 CURIOUS EXPLORATION

As shown in Fig.2(b), given the top-down gate formed by the principles $\mathcal{P}$ and the concept pool $\mathcal{C}$, CuES conducts bottom-up interaction with an *Explorer Agent* that, at each state $s_t$, receives only

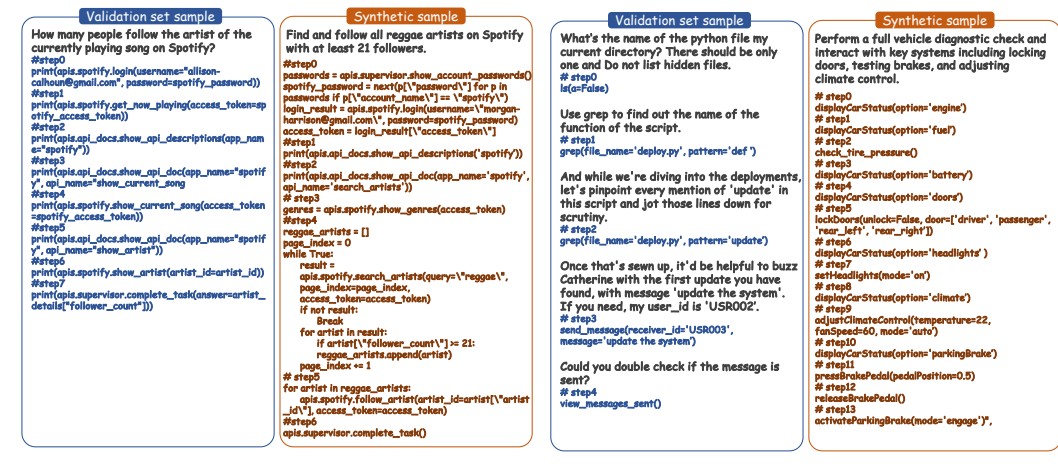

(a) AppWorld

(b) BFCL v3 Multi-Turn Base

Figure 3: Original (left) vs CuES-synthesized (right) for AppWorld and BFCL v3 Multi-Turn Base.

the currently executable candidate actions $\mathcal{A}(s_t)$, together with $\mathcal{P}$ and a small, randomly sampled subset of concepts $\mathcal{C}_t \subset \mathcal{C}$ (e.g., $|\mathcal{C}_t| = k$ drawn uniformly each rollout to promote coverage).

**Memory Tree** To enhance exploration diversity and avoid redundant traversal of the environment space, we introduce a **memory tree** $\mathbb{M}$. Indexed by environment states, the memory tree allows the Explorer to query the current state identifier $s_t$ and retrieve the set of actions $\mathbb{M}(s_t)$ that have previously been executed in states similar to $s_t$. Explorer then decides $a_t \in \mathcal{A}(s_t)$, with a preference for actions not yet attempted at local memory node ($a \notin \mathbb{M}(s_t)$), while taking into account priorities expressed by $\mathcal{P}$ and active concepts $\mathcal{C}_t$.

After executing $a_t$, the environment returns an observation $o_t$. The memory tree is updated at node $\phi(s_t)$ to record the attempt and its outcome via a compact sketch:

$$\mathbb{M}(s_t) \leftarrow \mathbb{M}(s_t) \cup \{a_t\} \quad (2)$$

The Explorer emits a triple per step,

$$z_t = (s_t, \ a_t, \ o_t), \quad (3)$$

and the exploration trajectory is the sequence $\mathcal{T} = \{z_t\}_{t=0}^{T}$, tagged with $(\mathcal{C}_t, \mathcal{P})$. These concept-aware, memory-informed trajectories are forwarded to the next stage in the pipeline (task abstraction), where low-level interactions are lifted into reusable task specifications while preserving the executability demonstrated during exploration.

## 3.3 Task Abstraction

Ref to Fig.2(c), this stage takes a *consecutive* mini-batch of exploration triples and lifts one or more multi-step tasks from it. Let a batch be

$$\mathcal{B} = \{ z_t = (s_t, a_t, o_t) \}_{t=t_0}^{t_0+B-1}, \qquad B = \texttt{batch\_size}, \quad (4)$$

where each triple consists of the step state $s_t$, the executed action $a_t$, and the resulting observation $o_t$. Rather than extracting query from every triple, we consider *contiguous segments* inside batch,

$$\mathcal{S} = \{ [i{:}j] \mid t_0 \leq i \leq j \leq t_0+B-1 \}, \quad (5)$$

and allow a single task to be supported by multiple triples.

For a candidate segment $[i{:}j] \in \mathcal{S}$, Task Agent extracts a new goal valid action sequence $g_{i:j}$. The executable query $q_{i:j}$ is a natural-language rendering of $g_{i:j}$, while the *guideline* is the action sequence $\text{guide}_{i:j}$) observed on the segment.

Preconditions are summarized from history $s_i$, and postconditions are given by $g_{i:j}$ holding at $o_j$. Each candidate $(q_{i:j}, \text{guide}_{i:j})$ receives a confidence $\sigma_{i:j} \in [0, 1]$ that reflects internal consistency

of the action–effect chain, clarity of the goal, and alignment with current principles and concepts. We keep those $\mathcal{Q}_{\mathrm{cand}}$ that pass the threshold:

To avoid redundancy, for each environment identifier $e = \mathrm{env\_id}$ we maintain a memory list of previously generated queries,

$$\mathbb{Q}_e = [\, q^{(1)}, q^{(2)}, \dots ]. \tag{6}$$

When proposing a new query, we form a context and supply $\mathcal{C}_e$ during query generation so that duplicates are explicitly flagged as "already generated."

## 3.4 QUALITY CONTROL

In Fig.2(d), this stage verifies the executability of each synthesized task using two agents. Let the input be the candidate set from task abstraction,

$$\mathbb{T} = \{\, (q_k, \ \mathrm{guide}_k, \ \sigma_k, \ e_k) \,\}_{k=1}^{N}, \tag{7}$$

where $q_k$ is a natural-language query, $\mathbb{T}$ means tasks, $\mathrm{guide}_k = [a_{i_k}, \dots, a_{j_k}]$ is the associated action guideline extracted from a contiguous segment, $\sigma_k$ is its confidence, and $e_k = \mathrm{env\_id}$ identifies the environment instance.

The Execution Agent receives $(q_k, \mathrm{guide}_k)$ and attempts to carry out the task in environment $e_k$, following the guideline actions in order while allowing minimal, environment-dependent deviations (for example, permission prompts). The attempt yields an execution trace

$$\tilde{\tau}_k = \big( (\tilde{s}_0, \tilde{a}_0, \tilde{o}_0), \dots, (\tilde{s}_{T_k}, \tilde{a}_{T_k}, \tilde{o}_{T_k}) \big), \tag{8}$$

with terminal observation $\tilde{o}_k^{\star} = \tilde{o}_{T_k}$. The trace, together with local outcomes (success messages, view changes), is returned for judgment.

Judge Agent checks both goal satisfaction and path faithfulness and verifies that observed actions respect the principles $\mathcal{P}$ and guideline up to allowable, environment-specific insertions (for example, a short detour to grant permission). A candidate is *accepted* if $reward = 1.0$ and *rejected* otherwise.

For accepted tasks, we log both the successful query and its execution trajectory as $\mathcal{T}_{\mathrm{synt}}$, so that subsequent stages and future explorations have access to validated problem statements and their witnessed solutions. Rejected candidates are filtered out and do not proceed further.

Table 1: Results on **WebShop LeaderBoard**.

| Model | Think | LLM | Params | greedy |
|---|---|---|---|---|
| qwen3-14b | ✗ | Qwen3 | 14B | 30.70% |
| qwen3-8b | ✗ | Qwen3 | 8B | 30.01% |
| qwen3-4b | ✗ | Qwen3 | 4B | 33.17% |
| qwen3-235b-a22b | ✗ | Qwen3 | 235B-A22B | 50.96% |
| deepseek-v3 | ✗ | Deepseek | / | 26.91% |
| qwen2.5-3b-instruct | ✗ | Qwen2.5 | 3B | 23.94% |
| qwen2.5-7b-instruct | ✗ | Qwen2.5 | 7B | 22.07% |
| qwen2.5-14b-instruct | ✗ | Qwen2.5 | 14B | 23.74% |
| **CuES (ours)** | ✗ | Qwen2.5 | 14B | **64.10%** |

Table 2: Results on **BFCL v3 Multi-Turn Base LeaderBoard**.

| Model | Think | LLM | Params | greedy |
|---|---|---|---|---|
| GPT-5 | ✗ | OpenAI | / | 33.5% |
| GPT-5-mini | ✗ | OpenAI | / | 31.5% |
| Gemini-2.5-Pro | ✔ | Gemini | / | 35.0% |
| Gemini-2.5-Flash | ✔ | Gemini | / | 36.0% |
| GPT-5-nano | ✗ | OpenAI | / | 33.5% |
| Amazon-Nova-Lite-v1:0 | ✗ | Amazon | / | 29.0% |
| o3 | ✔ | OpenAI o3 | / | 44.0% |
| DeepSeek-V3 | ✗ | DeepSeek | / | 43.5% |
| GPT-4o-mini | ✗ | OpenAI | / | 43.5% |
| Amazon-Nova-Pro-v1:0 | ✔ | Amazon | / | 42.5% |
| **CuES (ours)** | ✔ | Qwen2.5 | 14B | **44.2%** |

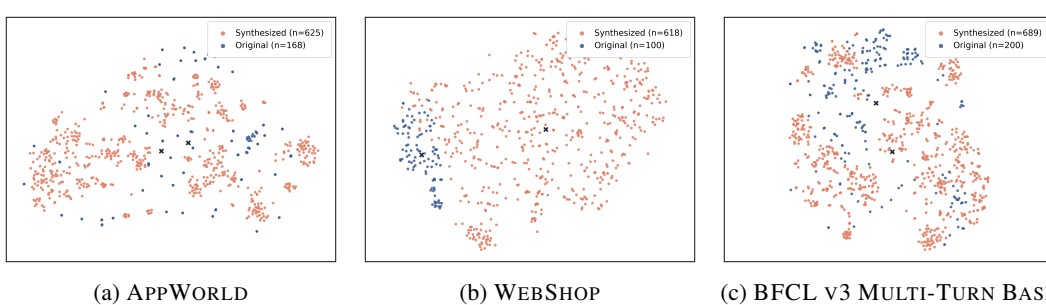

|              | (a) APPWORLD | (b) WEBSHOP | (c) BFCL V3 MULTI-TURN BASE |

Figure 4: Distribution comparison per environment in t-sne.

Table 3: Model performance (%) on three interactive environments using *avg@8* and *greedy*.

| Model | Params | AppWorld | | WebShop | | BFCLv3 | | Avg. | |
|---|---|---|---|---|---|---|---|---|---|
| | | avg@8 | greedy | avg@8 | greedy | avg@8 | greedy | avg@8 | greedy |
| Qwen2.5 | 3B | 0.00% | 0.23% | 20.65% | 23.94% | 6.94% | 7.00% | 9.20% | 10.39% |
| Qwen2.5 | 7B | 1.25% | 1.87% | 24.62% | 22.07% | 17.75% | 20.00% | 14.54% | 14.65% |
| Qwen2.5 | 14B | 11.76% | 14.29% | 25.74% | 23.74% | 25.69% | 31.50% | 21.06% | 23.18% |
| Qwen3 | 4B | 3.00% | 3.25% | 33.72% | 33.17% | 9.94% | 10.00% | 15.55% | 15.47% |
| Qwen3 | 8B | 17.76% | 21.43% | 26.92% | 30.01% | 18.75% | 18.50% | 17.81% | 19.98% |
| Qwen3 | 14B | 30.98% | 28.48% | 31.46% | 30.70% | 28.94% | 30.70% | 30.46% | 29.96% |
| **CuES (ours)** | 14B | **45.54%** | **45.24%** | **63.55%** | **64.10%** | **43.00%** | **44.15%** | **50.70%** | **51.16%** |

### 3.5 QUERY REWRITE

Ref to Fig.2(e), this stage operates only on the accepted set from quality control,

$$\mathcal{T}_{\text{synt}} = \left\{ (q_k, \text{ guide}_k, e_k, \tilde{\tau}_k) \right\}_{k=1}^{N}, \tag{9}$$

and produces a ladder of easier variants by progressively exposing elements of the guideline inside the query text while preserving goal semantics. For each tuple, let $\Gamma_k$ be a pool of rewrite hints distilled from $\text{guide}_k$ and the witnessed execution $\tilde{\tau}_k$ (action verbs, intermediate landmarks/views, parameter exemplars, and precondition CuES). We generate a sequence

$$q_k^{(0)} = q_k, \quad q_k^{(1)}, \ldots, q_k^{(L_k)}, \tag{10}$$

by iteratively injecting additional hints:

$$q_k^{(\ell+1)} = \text{AddHints}\big(q_k^{(\ell)}; \Delta\Gamma_k^{(\ell)}\big), \qquad \Delta\Gamma_k^{(\ell)} \subseteq \Gamma_k, \ \Delta\Gamma_k^{(\ell)} \neq \varnothing. \tag{11}$$

Rewriting stops when either target difficulty $D\big(q_k^{(\ell)}\big) \leq d_{\min}$ is reached or no further decrease is achievable.

## 4 EXPERIENCES

### 4.1 EXPERIMENTAL SETUP

We evaluate CuES on three interactive environments: AppWorld, WebShop, and BFCL v3 Multi-Turn Base. For each environment, we report the size of the original data and the amount synthesized by CuES under a matched interaction budget, alongside a brief description of the core task. Unless otherwise noted, we use the same runtime across environments: `model_name` = `qwen-plus-latest`, `temperature` = 0.7, `max_tokens` = 20480; Stage 1 rollout count = 500 with up to 30 steps per rollout; Stage 2 uses `batch_size` = 30 and `min_confidence` = 0.7; Stage 3 attempts up to 30 steps with `retry_attempts` = 3; the environment service has a 30s timeout and 30-step cap.

For benchmarks that include validation sets, such as BFCL v3 and WebShop, we use the validation set directly. For the BFCL v3, we use Multi-Turn Base setting as the test set. For AppWorld, we use test_normal set as the test set. The reported results are equivalent to the TGC indicator on the

Table 4: Synthesis-quality ablation with metrics.

| Setting | | | | AppWorld | | | | WebShop | | | |
|---|---|---|---|---|---|---|---|---|---|---|---|
| min_conf | batch | max_steps | Pool | PR | SR | ED | Step | PR | SR | ED | Step |
| 0.7 | 30 | 30 | ✗ | 0.6159 | 0.6477 | 0.0442 | 7.6500 | 0.1667 | 0.6780 | 0.1213 | 7.7812 |
| *Varying min_confidence (batch=30, max_steps=30, Req+Pool)* | | | | | | | | | | | |
| 0.5 | 30 | 30 | ✗ | 0.5868 | 0.6653 | 0.0450 | 7.8979 | 0.1481 | 0.6668 | 0.1143 | 8.6786 |
| 0.9 | 30 | 30 | ✗ | 0.6733 | 0.6622 | 0.0467 | 6.8415 | 0.1040 | 0.6772 | 0.1101 | 9.7619 |
| *Varying batch_size (min_conf=0.7, max_steps=30, Req+Pool)* | | | | | | | | | | | |
| 0.7 | 10 | 30 | ✗ | 0.6359 | **0.6903** | 0.0369 | 7.1637 | 0.1354 | **0.7240** | 0.0908 | 5.3191 |
| 0.7 | 50 | 30 | ✗ | **0.7239** | 0.6405 | 0.0385 | 7.7474 | 0.1064 | 0.6864 | 0.1129 | 7.6500 |
| *Varying max_steps (min_conf=0.7, batch=30, Req+Pool)* | | | | | | | | | | | |
| 0.7 | 30 | 20 | ✗ | 0.6439 | 0.6105 | 0.0480 | 7.2000 | 0.1196 | 0.6507 | 0.0899 | 4.9090 |
| 0.7 | 30 | 40 | ✗ | 0.6596 | 0.6136 | 0.0438 | **7.9741** | 0.1475 | 0.6897 | 0.0945 | **10.5938** |
| *Requirement / Concept Pool toggles (min_conf=0.7, batch=20, max_steps=30)* | | | | | | | | | | | |
| 0.7 | 30 | 30 | ✔ | 0.6642 | 0.6479 | **0.0506** | 7.7184 | **0.2111** | 0.5571 | **0.3429** | 7.3934 |

leaderboard. AppWorld has 715 tasks total—90 original and 625 CuES-synthesized. WebShop (689) and BFCL v3 (618) contain only CuES-synthesized data. In Fig.3, we also show samples of synthetic data and validation set data on the AppWorld and BFCL v3 datasets. It can be seen that the quality of our synthetic data is comparable to the original data, and we can synthesize more difficult data by adjusting the parameters.

## 4.2 Performance Comparison

We compare CuES with representative baselines(Grattafiori et al., 2024; Intelligence, 2024; Guo et al., 2024; Yao et al., 2023) on APPWORLD, WEBSHOP, and BFCL V3 MULTI-TURN BASE under two metrics: *avg@8* and *greedy*.

In Tab.6, Tab.1 and Tab.2, we compare CuES against recent baselines on AppWorld, WebShop, and BFCL v3 Multi-Turn Base. On AppWorld, CuES attains 45.24% greedy, surpassing the strongest listed baseline PlanExec with GPT-4o at 44.6%. The gain is most pronounced on WebShop, where CuES reaches 64.10% greedy versus 50.96% for qwen3-235b-a22b. On BFCL v3 Multi-Turn Base, CuES achieves 44.15% greedy, edging out o3 at 44.0%. Looking at Tab.3, CuES delivers a macro-average of roughly 51.2% greedy and 50.7% avg@8 across the three datasets, substantially higher than same-scale non-CuES backbones. These results indicate that environment-grounded synthesis with light top-down steering translates into consistent single-path and sampled execution gains across diverse domains. Notably, even when compared with models that have far more parameters or closed-source model than CuES (e.g., DeepSeek-V3(Liu et al., 2024), GPT-5, o3(Achiam et al., 2023) and Gemini-2.5 Pro(Comanici et al., 2025)), CuES still outperforms by a large margin, underscoring the impact of our synthesis rather than raw scale.

We also observe that CuES improves performance even under format shift on BFCL v3 Multi-Turn Base, where evaluation involves multi-turn follow-ups that do not strictly match the form of our synthesized queries. Despite this mismatch, CuES maintains a positive edge on greedy success, suggesting that the synthesized pool emphasizes executable, transferable intents rather than overfitting to any one prompt style. **Together, these findings support the claim that bottom-up exploration anchored by a concept pool and requirement confirmation can raise executability and coverage in ways that persist across backbones, scales, and interaction protocols.**

## 4.3 Hyperparameter Ablations and Distribution Analysis

For different benchmarks, we observed different changes in synthetic-quality with different settings. We believe that this may be closely related to the tasks. We selected the two benchmarks AppWorld and WebShop with the largest differences.

Ref to Tab.4, We assess synthesis quality with metrics after delexicalization and sentence embedding. Executability is measured by Pass Rate, diversity by Self-Redundancy over $k$ nearest neighbors, and distributional alignment by a relative energy distance between original and synthesized intent embeddings. The definitions are

$$\mathrm{PR} = \frac{|Q_{\mathrm{ok}}|}{|Q_{\mathrm{prop}}|}, \qquad \mathrm{SR@}k = \frac{1}{|Q_{\mathrm{ok}}|}\sum_i \frac{1}{k}\sum_{j\in\mathrm{kNN}(i)}\langle e_i, e_j\rangle, \qquad \mathrm{ED}_{\mathrm{rel}} = \frac{\mathrm{ED}(X,Y)}{\mathbb{E}_{i\neq i'}\|x_i - x_{i'}\|}. \quad (12)$$

**On APPWORLD, higher confidence improves executability with small costs to diversity and alignment.** Raising the threshold from 0.7 to 0.9 lifts PR from 0.6159 to 0.6733, while SR and ED change only slightly and the average step count falls from 7.65 to 6.84. Larger batches are effective: batch 50 reaches PR 0.7239 with controlled SR 0.6405 and ED 0.0385, outperforming batch 10 on PR while keeping alignment competitive. Extending the rollout depth from 20 to 40 steps increases PR to 0.6596 and reduces ED to 0.0438.

**WEBSHOP favors tighter alignment through smaller batches and shorter rollouts, while still gaining executability when needed.** With min_conf at 0.7, batch 10 improves ED to 0.0908 and reduces steps to 5.32 compared with batch 50, though PR is lower at 0.1354 versus 0.1064 for batch 50 the alignment gain is clear. Shorter rollouts also help alignment: max_steps 20 attains ED 0.0899 and Step 4.91, whereas max_steps 40 raises PR to 0.1475 with a mild ED increase to 0.0945 and a higher step cost. Enabling the concept pool strongly boosts PR from 0.1667 to 0.2111 and lowers SR from 0.6780 to 0.5571, yet shifts ED upward to 0.3429, reflecting targeted exploration toward salient subspaces.

## 4.4 VISUALIZATION OF SYNTHESIZED VS. ORIGINAL QUERIES

Fig.5 shows t-SNE projections of sentence embeddings for original and CuES-synthesized queries on AppWorld, WebShop, and BFCL v3. In AppWorld, the synthesized cloud envelops the sparse original points with nearly coincident centroids and small drift, indicating that CuES expands coverage around existing intent modes while remaining aligned. BFCL v3 exhibits the strongest agreement, with tightly interleaved clouds and very low relative energy distance, reflecting close intent matching. WebShop presents a deliberate shift: the synthesized cloud moves below the original cluster and $\mathrm{ED}_{\mathrm{rel}}$ increases, while self-redundancy decreases, revealing broader variety than the original set, which was concentrated in a narrow region. **Furthermore, we can observe that the distribution of synthetic data is more diverse than that of test data on all three datasets, which proves the diversity of synthetic data.**

**We complement the plots with concise qualitative samples. On both AppWorld and BFCL v3 benchmark, the synthetic sample showed better quality and diversity than the original sample.** Ref to Fig.3, in AppWorld, CuES surfaces executable workflows in the Spotify domain (login, enumerate genres, filter artists, follow), making affordances explicit and explaining the low drift and high pass rate. In BFCL v3, synthesized tasks articulate multi-system checks (vehicle states, controls, and brief terminal interactions), demonstrating precise tool grounding consistent with the tight embedding overlap.

## 5 CONCLUSION

We introduce CuES, a curiosity-guided, environment-grounded synthesis framework that unifies bottom-up exploration with top-down guidance to generate high-quality, executable agent data in query-free settings. Unlike prior methods that either decouple tasks from environment dynamics or drift under unconstrained exploration, CuES integrates lightweight top-down guidance—via requirement confirmation and concept pools—with bottom-up discovery powered by environment memory and intrinsic curiosity, **yielding diverse yet solvable trajectories while sharply reducing ineffective exploration.** Experiments on AppWorld, WebShop, and BFCL v3 show that CuES-synthesized data **not only match but surpass original datasets**, outperform strong baselines by an average of over 30 points on avg@8 and greedy metrics, and remain effective even against models with far larger parameter scales. These results demonstrate that curiosity-driven bottom-up synthesis, when coupled with minimal but strategic top-down control, tying executability, diversity, and distributional alignment to controllable hyperparameters, provides a scalable recipe for robust agent training in real environments. We view this top-down–meets–bottom-up design as a step toward reliable agent training corpora that adapt across domains and interaction protocols, and as a foundation for future on-policy synthesis and environment-specific reward modeling.

## ETHICS STATEMENT

This work does not involve human subjects, sensitive personal data, or any ethically concerning practices. All authors have read and adhered to the ICLR Code of Ethics, and explicitly acknowledge this during submission. Our study focuses exclusively on the algorithmic design and evaluation of a synthesis framework (CuES) in controlled, publicly available benchmarks: AppWorld, WebShop, and BFCL v3. These environments simulate interactions through APIs and toolchains rather than relying on real users, thereby avoiding privacy or security concerns. The data generated by our system consists of synthetic queries and trajectories automatically derived from environment dynamics and does not contain sensitive or proprietary content. Furthermore, our evaluation framework includes automatic quality control to filter out invalid or potentially harmful outputs, ensuring that the resulting corpora remain safe and aligned with community norms.

## REPRODUCIBILITY STATEMENT

We have taken extensive steps to ensure the reproducibility of our results. The main paper details the CuES pipeline and all hyperparameters, exploration settings, and evaluation protocols (Sec.3), while Appendix provides terminology clarifications and additional ablations. To further support reproducibility, we release partial source code, data preprocessing scripts, and configuration files, enabling others to replicate both the synthesis and evaluation stages. All experiments were conducted on public benchmarks (AppWorld, WebShop, and BFCL v3), which provide fixed environments, tool APIs, and task interfaces accessible to the community. Our reported results include averaged metrics (*greedy*, *avg@N*) with complete ablation studies across hyperparameters, ensuring transparency about performance variation. Together, these measures allow researchers to both verify our claims and extend CuES to new environments with minimal additional effort.

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

## A    DATASET STATISTICS

Table 5 summarizes original versus CuES-synthesized counts.

Table 5: Original vs. CuES-synthesized data counts across environments.

| Dataset | Orig. | Synth. | Total |
|---------|-------|--------|-------|
| AppWorld | 90 | 625 | 715 |
| WebShop | / | 689 | 689 |
| BFCL v3 | / | 618 | 618 |

## B    LEADERBOARD COMPARISON (APPWORLD)

Table 6: Results on **AppWorld LeaderBoard**.

| Model | Think | LLM | Params | greedy |
|-------|-------|-----|--------|--------|
| FullCodeRefl | ✗ | GPT-4o | / | 33.9% |
| FullCodeRefl | ✗ | GPT-4 Turbo | / | 25.6% |
| FullCodeRefl | ✗ | LLaMA3 | 70B | 24.4% |
| FullCodeRefl | ✗ | DeepSeekCoder | 33B | 13.1% |
| ReAct | ✗ | GPT-4 Turbo | / | 26.8% |
| ReAct | ✗ | LLaMA3 | 70B | 7.1% |
| ReAct | ✗ | DeepSeekCoder | 33B | 20.8% |
| IPFunCall | ✗ | GPT-4o | / | 32.1% |
| IPFunCall | ✗ | GPT-4 Turbo | / | 30.4% |
| PlanExec | ✗ | GPT-4o | / | 44.6% |
| PlanExec | ✗ | GPT-4 Turbo | / | 32.7% |
| PlanExec | ✗ | LLaMA3 | 70B | 8.9% |
| PlanExec | ✗ | DeepSeekCoder | 33B | 1.8% |
| **CuES (ours)** | ✗ | Qwen2.5 | 14B | **45.24%** |

## C    SIMILARITY ANALYSIS

Figure 5 visualizes the distribution of maximum cosine similarity between each synthesized query and the closest seed string. CuES preserves proximity to seeds in AppWorld and BFCL while expanding into novel regions; in WebShop it intentionally broadens coverage beyond a narrow seed cluster. And the overall distribution maintains a certain distance from the validation set.

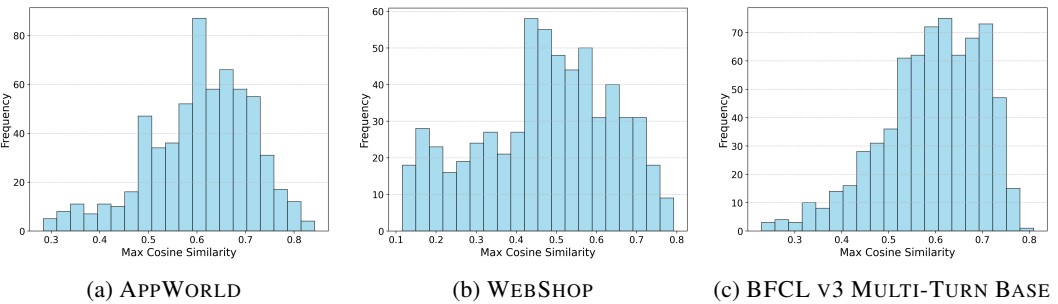

(a) APPWORLD          (b) WEBSHOP          (c) BFCL V3 MULTI-TURN BASE

Figure 5: Distribution of maximum cosine similarity between synthesized queries and any seed string, per environment.

## D    PROBLEM FORMULATION AND PIPELINE OVERVIEW

### D.1    GLOBAL OBJECTIVE

We formalize environment-grounded synthesis as learning a mapping from tools to executable tasks:

$$F : E \longrightarrow \mathcal{T}_{\text{synt}}, \tag{13}$$

where the environment is a set of tools $E = \{t_1, \ldots, t_m\}$ (e.g., APIs), and the output $\mathcal{T}_{\text{synt}}$ is a vetted set of tasks suitable for training. A *task* is a pair $(q, \tau)$ consisting of a natural-language *query* $q$ and

its executable *trajectory* $\tau = (s_1, a_1, o_1), \ldots, (s_n, a_n, o_n)$, a sequence of state–action–observation triplets. We also use an environment description $M$ (tool affordances, arguments, return types), an optional user need $U$, and an optional seed query set $S$ when available; when absent, the pipeline operates in a query-free mode.

## D.2  Five-Stage Functional Decomposition

To realize $F$ in equation 13, we compose five mappings that unify bottom-up discovery with top-down constraints, matching the notation used in Sec. 3.

**Stage 1: Requirement Confirmation ($F_{\text{req}}$).**  This stage summarizes the environment and derives top-down guidance:

$$F_{\text{req}} : E \mapsto M, \qquad \Pi : (U, M) \mapsto \mathcal{P}, \qquad \Phi : (M, S) \mapsto \tilde{\mathcal{C}}, \qquad \mathcal{C} = \text{Filter}_U(\tilde{\mathcal{C}}). \quad (14)$$

Here $M$ is the environment description, $\mathcal{P} = \{p_1, \ldots, p_r\}$ are actionable *principles* specifying format and priority actions, and $\mathcal{C}$ is the final *concept pool* obtained by filtering the preliminary pool $\tilde{\mathcal{C}}$ with $U$ when provided (otherwise $\mathcal{C} = \tilde{\mathcal{C}}$). $\Phi_{\text{env}}(M)$ and $\Phi_{\text{seed}}(S)$ are the environment- and seed-derived extractors implicit in $\Phi$.

**Stage 2: Curious Exploration ($F_{\text{explore}}$).**  Given $(M, \mathcal{P}, \mathcal{C})$, an Explorer interacts bottom-up while consulting an *environment memory* $\mathbb{M}$ to avoid redundancy:

$$F_{\text{explore}} : (E, M, \mathcal{P}, \mathcal{C}) \longrightarrow \mathcal{D}_{\text{traj}}, \qquad \mathcal{D}_{\text{traj}} = \left\{\tau^{(i)}\right\}_{i=1}^{N}, \ \tau^{(i)} = \left((s_t, a_t, o_t)\right)_{t=1}^{T_i}. \quad (15)$$

At each step $t$, the Explorer receives the admissible action set $\mathcal{A}(s_t)$, principles $\mathcal{P}$, and a random mini-pool $\mathcal{C}_t \subset \mathcal{C}$ to promote coverage, then emits $z_t = (s_t, a_t, o_t)$. The memory is updated to record attempted actions at the current state signature $\phi(s_t)$:

$$\mathbb{M}(\phi(s_t)) \leftarrow \mathbb{M}(\phi(s_t)) \cup \{a_t\}. \quad (16)$$

**Stage 3: Task Abstraction ($F_{\text{abstract}}$).**  Consecutive triples are lifted into candidate tasks by inducing a query and a concise guideline from contiguous segments:

$$F_{\text{abstract}} : \mathcal{D}_{\text{traj}} \longrightarrow \widetilde{\mathcal{T}} = \left\{(q, \text{guide}, \sigma, e)\right\}. \quad (17)$$

Here guide is the action sequence observed on the segment, $\sigma \in [0, 1]$ is a confidence score, and $e = \text{env\_id}$. To curb redundancy, for each $e$ we maintain a query list $\mathbb{Q}_e$ and supply it as context so that duplicates are explicitly flagged as "already generated."

**Stage 4: Quality Control ($F_{\text{qc}}$).**  Each candidate is executed and judged for solvability; only successful ones are retained:

$$F_{\text{qc}} : \widetilde{\mathcal{T}} \longrightarrow \mathcal{T}_{\text{synt}} \subseteq \widetilde{\mathcal{T}}. \quad (18)$$

The Execution Agent attempts $(q, \text{guide})$ in environment $e$ and returns a realized trace $\tilde{\tau}$; the Judge Agent verifies goal satisfaction and path faithfulness under $\mathcal{P}$.

**Stage 5: Query Rewrite ($F_{\text{rewrite}}$).**  Accepted tasks are reformulated by progressively exposing guideline hints to broaden coverage and lexical variety without relaxing executable intent:

$$F_{\text{rewrite}} : \mathcal{T}_{\text{synt}} \longrightarrow \mathcal{T}_{\text{synt}} \ (\text{enhanced}). \quad (19)$$

**End-to-end composition.**  The overall synthesis function is the composition

$$F = F_{\text{rewrite}} \circ F_{\text{qc}} \circ F_{\text{abstract}} \circ F_{\text{explore}} \circ F_{\text{req}}. \quad (20)$$

## D.3  Key Variables and Method-Agnostic Concepts

**States, actions, observations, and triplets.**  Each step is a triplet $z_t = (s_t, a_t, o_t)$ with $s_t \in \mathcal{S}$, $a_t \in \mathcal{A}(s_t)$, and $o_t$ the observation. A trajectory is $\tau = \{z_t\}_{t=1}^{T}$.

**Principles and concept pool.** $\mathcal{P} = \{p_1, \ldots, p_r\}$ are executable high-level rules (e.g., output schema, one-action-per-step, allowed verbs). $\mathcal{C} = \{c_1, \ldots, c_k\}$ is a pool of environment-specific concepts used as semantic seeds (e.g., in WebShop: "men's tops," "blue women's dress," "razor"), derived from $(M, S)$ and optionally filtered by $U$.

**Environment memory as a tree.** For an environment instance $e$, we view the memory as a directed tree $\mathbb{M}_e = (V_e, E_e)$ whose nodes summarize states via $\phi : \mathcal{S} \to \Sigma$ and whose edges are labeled by actions. The unseen frontier at node $\phi(s)$ is

$$\text{Frontier}_e(\phi(s)) \;=\; \mathcal{A}(s) \setminus \big\{\, a \,:\, (v \to v') \in E_e,\ \phi(s) = v,\ \lambda(v \to v') = a \,\big\}, \qquad (21)$$

prioritizing novel yet admissible actions during exploration.

**Queries, guidelines, and tasks.** A *query* $q$ is the natural-language problem; a *guideline* is the concrete action sequence extracted from trajectories; a *task* is $(q, \tau)$ where $\tau$ (or its guideline view) solves $q$ under $\mathcal{P}$ in environment $e$.

The separation between the *pipeline maps* in equation 14–equation 19 and the *problem variables* above mirrors Sec. 3, keeps symbols consistent with the method section $(M, U, S, \mathcal{P}, \mathcal{C}, \mathbb{M}, z_t, \tau)$, and makes the end-to-end objective $F : E \to \mathcal{T}_{\text{synt}}$ precise yet modular.

# E    ILLUSTRATIVE EXAMPLES OF CUES IN PRACTICE

**Requirement confirmation and concept pool construction (AppWorld, music domain).** In APPWORLD, the environment $M$ organizes many apps by category. Suppose the user intent is "explore music apps," i.e., $U$ targets the music category. We first build the preliminary pool $\tilde{\mathcal{C}} = \Phi_{\text{env}}(M) \cup \Phi_{\text{seed}}(S)$, where $\Phi_{\text{env}}(M)$ enumerates environment-defined entities and admissible actions (e.g., open, search, play, subscribe), and $\Phi_{\text{seed}}(S)$—if a seed query set is provided—parses noun phrases and action predicates such as *playlist*, *subscription*, and *cross-app comparison*. We then obtain the final pool $\mathcal{C}$ by filtering with the user need $U$, which retains only music-related apps, views, and actions. In parallel, we compute the principles $\mathcal{P}$, which are inserted into the next stage's prompts to prescribe output schema and emphasize priorities such as coverage and solvability within the music domain. If the user provides a seed query like "compare playlists across streaming apps," $\Phi_{\text{seed}}(S)$ contributes additional concept atoms that survive the filter when consistent with $U$. If $U$ is absent but $S$ is present, $\mathcal{C}$ defaults to the unfiltered $\tilde{\mathcal{C}}$, enabling seed-informed yet query-free exploration. If neither $U$ nor $S$ is provided, $\mathcal{C}$ reduces to the environment-wide inventory $\Phi_{\text{env}}(M)$ and $\mathcal{P}$ becomes neutral, allowing unbiased, seedless discovery.

**Task abstraction from consecutive triples.** In APPWORLD, a consecutive batch may contain triples that collectively move from an app home to a playlist view within a music app. A selected segment $[i{:}j]$ yields a goal such as "reach the playlist page," a query $q_{i:j}$ expressing that goal, and a guideline equal to the observed actions on that segment. If $\mathbb{Q}_e$ already contains an equivalent query, $\mathcal{C}_e$ lists it as "already generated," and the candidate is discarded. The resulting $\mathbb{Q}_e$ is forwarded to the quality control stage in the pipeline.

**Difficulty ladder via query rewriting.** Consider a music-domain guideline observed on Spotify. A single validated task can yield a difficulty ladder by gradually revealing guideline content:

- Hardest: "Find the most played song of artist *A*."

- Medium: "On Spotify, find the most played song of artist *A*."

- Easiest: "If needed, retrieve Spotify credentials; open Spotify; search for artist *A*; sort results by play count; open the top result."

These variants share the same validated goal but differ in how much of the guideline is surfaced in the query, which calibrates difficulty while maintaining executability.

## F STABILITY OF SYNTHESIZED DATA

To study the effect of exploratory randomness in the synthesis stage, we ran CuES three times on AppWorld under the same synthesis configuration, and then trained a separate Qwen2.5-14B-Instruct agent on each synthesized dataset.

The resulting avg@8 scores on AppWorld were

$$43.21, \ 46.67, \ 45.14 \quad (\text{mean } 45.01, \text{ std } 1.73),$$

all exceeding a 30-point gain over the baseline (11.76%). This shows that despite stochastic exploration, CuES consistently produces high-quality training data and that downstream performance is stable across independently synthesized datasets.

## G SYNTHESIS COST AND EFFICIENCY

We provide a more concrete cost profile per accepted task. Under the configuration rollout $= 50$, max_turn $= 30$ (i.e., 1,500 environment steps per run) and task abstraction batch size $= 30$, we ran CuES three times on AppWorld. Table 7 reports abstraction, execution, and rewriting statistics.

Table 7: Synthesis cost under rollout $= 50$, max_turn $= 30$ on AppWorld.

| Run | Abs. | Succ. | Succ.% | Rew. | Exp.$\times$ | Step/Succ. | Step/Final |
|---|---|---|---|---|---|---|---|
| #1 | 125 | 69 | 55.2 | 182 | 2.64 | 21.7 | 8.2 |
| #2 | 150 | 78 | 52.0 | 212 | 2.72 | 19.2 | 7.1 |
| #3 | 132 | 71 | 53.8 | 204 | 2.87 | 21.1 | 7.4 |
| avg.$\pm$std. | 135.7$\pm$12.8 | – | 53.7$\pm$1.6 | – | 2.74$\pm$0.12 | 20.7$\pm$1.3 | 7.6$\pm$0.6 |

On average, CuES produces about 136 abstracted tasks per 1,500 environment steps, with a success rate of 53.7% and a rewriting expansion factor of 2.74$\times$. This corresponds to roughly 7.6 environment steps per kept final task. In the revised version, we additionally provide a precision–recall versus budget curve to show how acceptance rates saturate as rollout depth increases, allowing practitioners to choose a practical operating point.

## H BEHAVIORAL COVERAGE

We analyzed behavior-level coverage on AppWorld with a focus on tool usage.

Across random initializations, each AppWorld environment instance exposes roughly 20–40 tools. Evaluation episodes typically involve 2–10 tools, while CuES-synthesized tasks involve approximately 4–20 tools per trajectory (non-exhaustive but consistently higher coverage than the eval set). This is consistent with the average solution length of around 10 steps reported in the main text: by relaxing filters and using deeper rollouts, CuES naturally explores more tool combinations and long-tail affordances.

## I ROBUSTNESS OF THE LLM JUDGE

CuES relies on an LLM-based judge to verify goal satisfaction and trajectory faithfulness for synthesized tasks. To assess its reliability, we used the AppWorld dev split, where an official environment reward is available. For each rollout, we computed both the environment's native success signal and the LLM judge's decision, and compared them.

We observed an order-consistency ratio of 0.8345 between the two evaluations, with Pearson correlation 0.6921 and Spearman correlation 0.6143. These values indicate a strong positive alignment between the LLM judge and the ground-truth environment reward.

## J    CROSS-DOMAIN TRANSFER

To probe whether CuES learns environment-specific tricks or more general interaction patterns, we conducted cross-domain transfer experiments between AppWorld and BFCL. We trained an agent on one benchmark using CuES data and evaluated it on both benchmarks without additional finetuning.

Table 8: Cross-domain transfer between AppWorld and BFCL (avg@8).

| Training Domain | AppWorld | BFCL |
|---|---|---|
| Baseline (no CuES) | 11.76 | 25.74 |
| CuES on AppWorld | 45.54 | 36.43 |
| CuES on BFCL | 20.18 | 43.00 |

As shown in Table 8, training on CuES data from one domain significantly improves performance on the other, even though AppWorld and BFCL differ substantially in available tools and task templates. This supports the claim that CuES captures transferable interaction structures rather than overfitting to a single benchmark's idiosyncrasies.

## K    RL TRAINING CONFIGURATION

For completeness, we summarize the RL training configuration that was only sketched in the main text. For each benchmark (AppWorld, BFCL, WebShop), we train a separate Qwen2.5-14B-Instruct agent using a GRPO-style objective, without mixing data across environments. We use a global train batch size of 16, a maximum prompt length of 4096 tokens, a maximum response length of 20480 tokens, AdamW optimizer with learning rate $1 \times 10^{-6}$, and 30 epochs with periodic validation and checkpointing. For original benchmark data, the reward is the environment's native pass/fail signal; for CuES-synthesized data, the reward comes from the LLM judge described above.

## L    THE USE OF LLMS

LLMs were not used in the research ideation or as contributors to experimental results. Their role was limited to proofreading: we employed LLM-based tools to check grammar and spelling errors in the writing stage. All substantive research contributions, including framework design, implementation, and evaluation, were made solely by the authors.

