# OpenReview forum: "CUES: Bottom-Up Exploration and Top-Down Guidance for Agentic Data Synthesis"
_ICLR.cc/2026/Conference — Submitted to ICLR 2026_

### Official Review · Reviewer_GfAy · 2025-10-24

**Soundness:** 3
**Presentation:** 2
**Contribution:** 3
**Rating:** 4
**Confidence:** 3

**Summary:**

CuES is a framework for generating training data for LLM agents that combines bottom-up exploration with minimal top-down guidance. It explores an environment to identify solvable interaction patterns, converts them into tasks, verifies executability, and rewrites queries to adjust difficulty—without requiring seed queries. Two simple controls—an Environment Memory and Concept Pools—keep exploration grounded and aligned with user goals. On AppWorld, BFCL, and WebShop, CuES produces diverse, executable tasks and improves downstream RL performance.

**Strengths:**

1. CuES blends bottom-up task discovery with two light top-down controls. Environment Memory helps revisit useful states, and Concept Pools steer curiosity toward goals without scripting solutions. This keeps the system grounded yet focused.

2. The pipeline does not rely on seed queries or external corpora, and it validates each candidate by checking goal satisfaction and trajectory faithfulness before keeping it. Only verified successes are logged for later stages.

3. On AppWorld, WebShop, and BFCL, CuES yields data that matches or exceeds curated sets and improves downstream training.

4. The framework is designed to produce executable, varied tasks, and analyses show broad coverage while maintaining distance from validation sets. Dataset statistics also indicate a substantial synthesized scale across environments.

**Weaknesses:**

1. The setup fixes substantial budgets (e.g., 500 rollouts, up to 30 steps; long token limits) but does not report synthesis cost per accepted task (tokens, wall-clock) or rejection rates beyond PR. Please include cost-per-kept-query, attempts-per-accept, and an efficiency curve (PR vs. budget) so others can plan deployments.

2. The paper relies on SR@k and relative energy distance computed in embedding space plus t-SNE visuals; these capture distribution shape but can miss behavioral coverage (skills, tools, preconditions). Add behavior-level measures like API/tool coverage, unique action schemas, success-path novelty, and lexical overlap checks against eval sets to rule out near-duplicates.

3. The table toggles the concept pool and shows a trade-off (PR rises but ED increases on WebShop), yet there’s no analogous on/off ablation for the Environment Memory to quantify its contribution. Add a memory ablation and a sweep over pool strength/selection size to map the diversity–executability frontier, and report downstream RL sensitivity to these knobs. This would clarify how much each top-down cue drives gains vs. side effects.

4. Acceptance hinges on a judge verifying goal satisfaction and path faithfulness, with a binary reward=1.0 criterion. It would help to stress-test this with adversarial or ambiguous tasks (e.g., partial success, detours) and to report false-accept/false-reject rates, perhaps via a small human audit or secondary checker.

**Questions:**

1. When I click the anonymous GitHub link, it shows "Page not found." Is there an issue?

2. How did you do the RL training? Can you provide more details?

3. How sensitive are your gains to the strength and composition of the Concept Pool and the mini-pool sampling during exploration? Could you share ablations sweeping pool size/filters and the Environment Memory settings?

---

> ### Author Response · Authors · 2025-11-16
> **Thanks to reviewer GfAy for your careful review and time.**
>
> **Thank you very much for your time and careful review.** It is clear from your comments that you are very familiar with agentic RL and data-centric frameworks, and your feedback helped us realize several aspects that were under-explained in the initial submission. We address your concerns point by point below.
>
> 1. Budgets and synthesis cost.
>
>  Under the configuration (rollout = 50, max_turn = 30 → 1,500 env steps), we ran three independent synthesis runs on AppWorld. The statistics are:
>
> | Run | Abs. | Succ. | Succ.% | Rew. | Exp.× | Step/Succ. | Step/Final |
> |:---:|:----:|:----:|:------:|:----:|:----:|:----------:|:----------:|
> | #1 | 125 | 69 | 55.2 | 182 | 2.64 | 21.7 | 8.2 |
> | #2 | 150 | 78 | 52.0 | 212 | 2.72 | 19.2 | 7.1 |
> | #3 | 132 | 71 | 53.8 | 204 | 2.87 | 21.1 | 7.4 |
> | avg.±std. | 135.7±12.8 | – | 53.7±1.6 | – | 2.74±0.12 | 20.7±1.3 | 7.6±0.6 |
>
> **Aggregated, this corresponds to ≈ 2.7× expansion, ≈ 54% execution success, and ≈ 7.6 env steps per kept query under a 1,500-step budget.**
>
> 2. Behavioral coverage beyond SR/ED.
>
>  We agree that SR@k and embedding-based ED mainly capture distribution shape. To better assess behavioral coverage, we computed tool coverage on AppWorld. Across random initializations, the environment exposes roughly 20–40 tools; evaluation rollout typically involve 2–10 tools, while CuES-synthesized tasks involve 4–20 tools per rollout (rough, but consistently higher than the eval set). This aligns with the trajectory statistics in the main text: the average solution length is around 10 steps, and by **adjusting filtering and exploration hyperparameters we can trade more steps for higher tool coverage.**
>
> 3. Environment Memory ablation.
>
> We agree that memory deserves a clearer ablation. In early experiments without Environment Memory and Task Memory, our self-redundancy (SD) metric **fluctuated around 0.74, noticeably higher than the current 0.60–0.70 range.** This indicates that, without memory, exploration spends more budget on redundant loops and produces more internally similar trajectories, whereas with memory the same budget yields more diverse and non-degenerate behaviors. We will consider adding experiments to the main text.
>
> 4. Judge reliability.
>
> To stress-test the LLM judge, we used the AppWorld dev set where an official reward is available. We rollout trajectories, evaluated them both with the environment’s native success signal and with our LLM judge, and compared the two. **We observed an order-consistency ratio of 0.8345 between the two evaluations, with Pearson correlation 0.6921 and Spearman correlation 0.6143, indicating a strong positive alignment between the LLM judge and the ground-truth environment reward.** This suggests that LLM judge reward is reliable in practice.
>
> 5. GitHub link.
>
> Thank you for flagging the repository issue. The error is caused by an extra “.” at the end of the URL. We will fix this typo.
>
> 6. RL training details.
>
> We acknowledge that some experimental details were compressed in the main text due to space limits and will move a concise but complete description to the appendix. Our RL training follows a standard GRPO-style setup. For each benchmark, we train a separate Qwen2.5-14B-Instruct agent, without mixing data across environments. We use a global batch size of 16, max prompt length 4096, max response length 20480, AdamW with learning rate 1e-6, and 30 epochs with periodic validation. For original benchmark data, the reward is the environment’s own pass/fail signal; for CuES-synthesized data, the reward comes from the LLM judge as described above.
>
> 7. Concept Pool strength and mini-pool sensitivity.
>
> For fairness, we did not use CP in the main AppWorld and BFCL experiments reported in the paper, since CP can inject prior knowledge that might confound comparisons. In early experiments and in WebShop, we did use CP: the total CP size is derived from high-level information (e.g., seed descriptions or categories), and the mini-pool size during exploration is typically 8–16. **Within this range, the gains from CuES are stable; extremely small or very large pools lead to the expected trade-off between diversity and executability.**
>
> **Your review is very valuable to our work.** The additional analyses, clarifications, and experiments discussed in this response will be incorporated into the main paper or appendix. Once again, thank you for your careful and technically grounded review. If these clarifications resolve your concerns, we would be grateful if you could consider raising your score.

---

> ### Author Response · Authors · 2025-11-24
> **With about a week left in the discussion period, we remain ready to clarify any questions or concerns.**
>
> Thanks for the constructive, deployment-oriented questions. We **added synthesis efficiency statistics** using three AppWorld runs under fixed budgets, reporting steps per successful task and per final kept task. For behavioral coverage, we **provided tool-coverage analysis** showing synthesized tasks activate more tools per trajectory than eval tasks, with coverage increasing smoothly under deeper rollouts/looser filters. We also **included early Environment Memory ablations** showing reduced self-redundancy when memory is enabled. **Judge reliability** was stress-tested by comparing LLM-judge decisions to AppWorld’s native rewards, showing high agreement and strong Pearson/Spearman correlations. Finally, we fixed the anonymous GitHub link issue (**a trailing “.” typo**) and summarized RL + CP sensitivity details, noting CP was disabled for fairness in AppWorld/BFCL main results.
>
> **We note that the author–reviewer discussion period has about one week remaining.** If these clarifications resolve your concerns (or if any points still need further evidence), we would greatly appreciate your brief follow-up so we can address remaining issues before the discussion window closes. If these clarifications resolve your concerns, we would be grateful if you could consider raising your score.

---

> ### Author Response · Authors · 2025-11-27
> **Since the discussion period ends in less than 5 days, we wanted to check in to make sure we have addressed your concerns as clearly as possible.**
>
> Thank you for your detailed and technically grounded review, and for highlighting concrete aspects that would improve the paper’s practicality. Since the discussion period ends in **less than 5 days**, we wanted to check in to make sure we have addressed your concerns as clearly as possible.
>
> If there are remaining points you feel are not fully resolved, especially around efficiency curves vs. budget, memory/CP sweeps, or judge false-accept/false-reject characterization, please let us know and we will address them directly. If our rebuttal has clarified your concerns, we would sincerely appreciate it if you would consider **raising your rating**.
>
> Thank you again for your time and for helping us strengthen the work.

---

### Official Review · Reviewer_gNE7 · 2025-10-26

**Soundness:** 3
**Presentation:** 2
**Contribution:** 2
**Rating:** 4
**Confidence:** 4

**Summary:**

The paper introduces CuES, a query-free framework for synthesizing high-quality, executable training tasks for LLM-based agents by unifying curiosity-driven bottom-up exploration with lightweight top-down guidance. CuES runs in five stages—requirement confirmation, curious exploration, task abstraction, quality control, and query rewriting—using a concept pool and environment memory to steer exploration toward valid affordances, and explicit execution/judging to ensure tasks are solvable. It generates diverse, environment-grounded tasks without manual seed queries, then adjusts difficulty via progressive hint injection. Experiments on AppWorld, WebShop, and BFCL v3 show CuES’s synthesized data match or surpass original datasets and yield strong downstream performance, outperforming larger or closed-source baselines (e.g., 64.1% greedy on WebShop and ~45% on AppWorld/BFCL). Ablations demonstrate controllable trade-offs between executability, diversity, and alignment via confidence thresholds, batch size, rollout depth, and concept pools. Overall, CuES bridges top-down goal clarity with bottom-up grounding, providing a scalable recipe for cost-effective agent training in complex environments.

**Strengths:**

1. The proposed method has some design in task abstraction and quality control in data synthesis.
2. The results can surpass strong proprietary models like GPT-5, but more training details need to be confirmed.

**Weaknesses:**

1. Some experiment settings are not clear: (a). Do you use the user need and seed query to synthesize data? Section 3.1 only says they are optional, but I would like to know if you use them in experiments. (b). The training configuration is missing: Do you mix the data synthesized across benchmarks to train a model or train one model for each benchmark? Training steps, batch size, learning rate, loss, accuracy across checkpoints and other dynamics are also missing. (c). The RL setup, algorithm, rewards, etc. are also missing.
2. From the Figure 3, the synthesized data looks similar to the validation set (maybe more difficult). Training on such data poses the risk of overfitting.

**Questions:**

1. typo in line 057: ... feasible in situ
2. typo in line 060: ... only targets -> only target
3. There is a missing comparison with Learn-by-interact (https://arxiv.org/pdf/2501.10893). This work uses backward construction to align with goals, synthesizes subtasks that could be applied in different scenarios, and requires LLMs to follow instructions in exploration in the data synthesis to improve efficiency. This contradicts the descriptions in line 056-061.
4. Why there is no thinking in Table 1 with open-sourced models, but there is thinking in Table 2 with proprietary models.

---

> ### Author Response · Authors · 2025-11-16
> **Thanks to reviewer gNE7 for your technically review and time.**
>
> **Thank you very much for your time and careful review.** We address your concerns point by point below.
>
> 1. Experiment settings.
>
> We acknowledge that some experimental details were compressed in the main text due to space limits and will add to the appendix.
>
> (a) Requirement Confirmation (RC) and Concept Pools (CP).
>
> RC and CP are optional, soft top-down controls. In the results on AppWorld and BFCL, we **do not use RC/CP**, so tasks are discovered purely bottom-up. On WebShop, search APIs require an input category; we therefore initialize a coarse CP entry (e.g., “electronics”). In early experiments, enabling RC/CP under the same rollout budget improved avg@8 by 5–7 points, so them are **useful and optional**.
>
> (b) Training configuration.
>
> We **train one independent agent per benchmark.** For each of AppWorld, BFCL, and WebShop, we fine-tune Qwen2.5-14B-Instruct with a GRPO algo., using a global batch size of 16, max prompt length 4096, max response length 20480, AdamW with lr=1e-6, and 30 training epochs. The full configuration will be summarized in the appendix.
>
> (c) RL setup and rewards.
>
> Our RL setup follows a standard GRPO algo.. For original benchmark data, we use the native success signal. For CuES-synthesized data, we use an LLM judge guided by CuES guidelines to provide sparse rewards. We **do not share synthesized data across benchmarks.**
>
> 2. Risk of overfitting.
>
> We understand the concern that synthesized trajectories look similar to test cases. Conceptually, CuES **only interacts with environment APIs and never sees test labels or task lists.** It observes states, actions, and transitions, but not ground-truth evaluation annotations. Thus, it is natural that synthesized data are semantically close to validation examples (same environment distribution), but they are not copies of held-out tasks.
> Empirically, two observations suggest that **CuES is not simply memorizing specific instances.** First, different synthesis runs with different seeds produce different datasets but consistently strong training performance, indicating stability under synthesis randomness. Second, cross-domain transfer shows that CuES learns domain-agnostic interaction patterns:
>
> | Training Domain | AppWorld |  BFCL |
> |:---------------:|:----------------:|:-------------:|
> | **Baseline** | 11.76% | 25.74% |
> | **AppWorld** | **45.54%** | **36.43%** |
> | **BFCL** | **20.18%** | **43.00%** |
>
> Training on CuES data from one environment significantly improves performance on another with different tools and task structures.  Together, this supports that CuES captures reusable structural patterns rather than overfitting to visually similar examples.
>
> 3. Typos.
>
>  We will correct.
>
> 4. Comparison with Learn-by-interact.
>
>  We appreciate the pointer to LBI and agree it is an important related framework. Both LBI and CuES are data-centric and synthesize agent–environment trajectories, but they differ in how tasks and guidance are defined. **LBI is instruction-first and documentation-driven**: it uses self-instruct over external resources (documentation, tutorials) to generate task instructions, then executes them and applies backward construction to align instructions and trajectories and to build retrieval for ICL and training. In contrast, CuES is curiosity-driven and seed-free at its core: it **does not rely on predefined seed queries or external corpora,** but discovers tasks bottom-up from raw interaction traces through a five-stage pipeline (RC, curiosity-guided exploration, Task Abstraction, Quality Control, Query Rewriting). On top of this bottom-up backbone, CuES adds lightweight top-down mechanisms: Environment Memory records salient states for revisitation, and CP softly conditions exploration on training objectives to guide curiosity and improve efficiency, without prescribing concrete tasks. We will revise lines 56–61 and the related work section to explicitly cite LBI and clarify that our description contrasts instruction-driven, external-resource-based pipelines with CuES’s environment-grounded, seed-free but lightly guided design.
>
> 5. “Thinking” in Table 1 vs Table 2.
>
>  We apologize that the difference between Table 1 and Table 2 was not clearly explained. WebShop **has no official leaderboard**, so all numbers in Table 1 are from our own unified evaluation pipeline. BFCL **has an official leaderboard**. When we re-ran proprietary baselines under our unified pipeline, their performance was lower than the leaderboard numbers, mainly due to prompt and config differences. To avoid underestimating baselines, we adopt the best official leaderboard results in Table 2. This is why “thinking” explicitly appears in Table 2 but not in Table 1.
>
> Once again, **thank you for your careful and technically grounded review.** The additional analyses in this response will be incorporated into the main paper or appendix.  If these clarifications resolve your concerns, we would be grateful if you could consider raising your score.

---

> ### Author Response · Authors · 2025-11-24
> **With about a week left in the discussion period, we remain ready to clarify any questions or concerns.**
>
> We appreciate your technically grounded feedback. We clarified all missing experimental settings: RC/CP usage (**disabled for AppWorld/BFCL main results,** coarse CP only for WebShop due to API requirements), one agent trained per benchmark without cross-mixing, and a concise GRPO training + reward specification. The overfitting concern is addressed both conceptually (**CuES cannot access held-out task lists/labels**) and empirically via multi-run synthesis stability and cross-domain transfer. We also added an explicit comparison to Learn-by-interact and **explained why “thinking”** appears in BFCL leaderboard baselines but not in our WebShop runs (budgeted non-thinking mode, with thinking-mode results being added).
>
> **We note that the author–reviewer discussion period has about one week remaining.** If these clarifications resolve your concerns (or if any points still need further evidence), we would greatly appreciate your brief follow-up so we can address remaining issues before the discussion window closes. If these clarifications resolve your concerns, we would be grateful if you could consider raising your score.

---

> ### Comment · Reviewer_gNE7 · 2025-11-25
> **Thank you for the response**
>
> Thanks a lot for the additional analysis and results! It is interesting to see that training on AppWorld significantly improves results in BFCL. Could you provide more insights on why this could happen? Ideally, generalization usually comes from diverse training data. It  would be more helpful to see analysis/examples to understand the ideas behind results.

---

> > ### Author Response · Authors · 2025-11-26
> > **Thank you for the thoughtful follow-up.**
> >
> > Thank you for the thoughtful follow-up. We agree that cross-domain generalization often benefits from **diverse** training data. In agentic RL / tool-use settings, however, diversity should be understood not only as topical/textual diversity, but more importantly as **diversity of executable interaction patterns**, i.e., whether training trajectories cover transferable **tool-use primitives**. We believe the performance arise exactly from this kind of interaction diversity. We will use the migration from Appworld to BFCL as an example to explain the underlying principles.
> >
> > 1. Mechanistic insights
> >
> > a. What transfers is *tool-use skill primitives*, not app-specific knowledge
> > AppWorld and BFCL differ in surface domain (apps/scenarios), but they are highly aligned in the underlying requirement: **closed-loop decision making under tool constraints**. AppWorld training repeatedly reinforces the following capabilities that BFCL also critically tests:
> >
> > - **(P1) Tool/Schema grounding:** selecting actions strictly from the provided tool list and respecting the API signatures.
> > - **(P2) Constraint satisfaction:** compiling natural-language constraints into executable checks/branches (e.g., thresholds, prerequisites).
> > - **(P3) Error-driven recovery:** interpreting environment/tool errors, adding missing prerequisite actions, and retrying correctly instead of oscillating among invalid calls.
> > - **(P4) Compositional tool chaining:** completing a task by composing multiple tools and reusing intermediate states across subsequent user requests.
> >
> > These primitives are domain-agnostic and therefore transfer well from AppWorld to BFCL even when the specific tools differ.
> >
> > b. AppWorld provides denser, more actionable feedback, which trains **closed-loop** policies
> > Compared to many static function-call datasets, AppWorld interaction yields structured feedback. Under RL, such feedback encourages a policy pattern of:
> > >observe tool feedback → revise the next step → re-verify
> >
> > This closed-loop behavior directly matches BFCL’s multi-step tool calling requirements and improves robustness.
> >
> > 2. Evidence via examples
> >
> > Below we provide a compact behavioral contrast, summarizing the two examples. We omit long context and keep only the key actions that illustrate why transfer occurs.
> >
> > a. Untrained baseline
> >
> > A common failure pattern is repeated invocation of **non-existent APIs** and proceeding via assumptions rather than re-grounding to the available tool/schema.
> >
> > Goal: download liked songs from Spotify album library.
> >
> > 1) login -> obtain access_token
> > 2) call spotify.get_liked_songs(...)              -> ERROR: No such API
> > 3) switch to spotify.get_current_user_saved_tracks(...) -> ERROR: No such API
> > 4) switch to spotify.get_user_saved_tracks(...)   -> ERROR: No such API
> > 5) repeats oscillation among non-existent APIs
> > 6) eventually: "assume the necessary APIs will be available" (ungrounded)
> >
> > this is not merely **domain knowledge missing**; it reflects **missing (P1) schema grounding and (P3) error-driven recovery.** The policy does not return to the current tool list/specification to re-plan; instead it explores an invalid action space.
> >
> > b. AppWorld-trained model
> >
> > In contrast, the AppWorld-trained model exhibits the transferable primitives that BFCL rewards: read error → add missing prerequisite → retry; implement conditional logic from the user request; and chain across tools.
> >
> > User intents:
> > (1) secure car (lock doors, engage parking brake),
> > (2) start engine to monitor fuel/battery,
> > (3) check tires; if any < 40 PSI -> find nearest tire shop,
> > (4) tweet update with hashtag/mention, then comment.
> >
> > Core trace:
> > - lockDoors(unlock=false, door=[driver, passenger, rear_left, rear_right])
> > - activateParkingBrake(mode=engage)
> > - startEngine(START) -> ERROR: "Brake pedal needs to be pressed"
> >   => pressBrakePedal(pedalPosition=1.0)
> >   => startEngine(START) -> engineState=running
> > - displayCarStatus(fuel); displayCarStatus(battery)
> > - check_tire_pressure() -> pressures in 32~35 PSI
> >   IF min_pressure < 40:
> >      find_nearest_tire_shop() -> returns location
> > - post_tweet(content="Tires checked and engine purring smoothly!",
> >             tags=["#RoadTrip"], mentions=["@AutoUpdates"])
> > - comment(tweet_id=10,
> >           comment_content="Safety first! Remember tire checks are crucial.")
> >
> > the improvement aligns with the same primitives BFCL evaluates:
> >
> > (P3) Error-driven recovery (engine start requires pressing brake; the model fixes and retries).
> >
> > (P2) Constraint satisfaction (explicit threshold-based branching for tire pressure).
> >
> > (P4) Compositional chaining across tools (vehicle checks → location lookup → Twitter post → comment).
> >
> > Overall, we believe the observed AppWorld→BFCL gains are best explained by **training robust, transferable tool-conditional closed-loop policies, rather than memorizing domain-specific content.** If these clarifications resolve your concerns, we would be grateful if you could consider raising your score.

---

> ### Comment · Reviewer_gNE7 · 2025-11-26
> **Thank you**
>
> Thanks a lot for the explanation! My concerns are fully addressed and I have updated my rating accordingly!

---

### Official Review · Reviewer_Y3MX · 2025-10-31

**Soundness:** 3
**Presentation:** 3
**Contribution:** 3
**Rating:** 4
**Confidence:** 4

**Summary:**

The paper proposes CuES (Curiosity-driven and Environment-grounded data Synthesis), a framework that removes reliance on manual seed queries, discovers tasks bottom-up from interaction traces, and then explicitly executes and judges candidates before query rewriting. Two lightweight top-down controls—Requirement Confirmation and Concept Pools—shape coverage and reduce wasted exploration without prescribing exact tasks; an Environment Memory stores compact state–action sketches to revisit informative frontiers and avoid redundant loops. The pipeline has five stages: requirement confirmation, curiosity-guided exploration, task abstraction, quality control, and query rewriting. The authors evaluate CuES on AppWorld, BFCL, and WebShop, reporting synthesized data that match or surpass original datasets and lead to strong downstream training performance.

**Strengths:**

1. The motivation is clear. The paper contrasts top-down imitation (seed queries/LLM expansion) with bottom-up exploration (derive tasks from interactions), arguing CuES integrates the clarity of the former with the executability of the latter.
2. The proposed framework is lightweight and efficient. Requirement Confirmation and Concept Pools “soft-aim” exploration without scripting solutions; Environment Memory mitigates redundancy by revisiting salient states. This balances diversity with usable coverage.
3. This paper uses a series of experiments to confirm the performance. On AppWorld/BFCL/WebShop, synthesized data reportedly match or exceed original datasets, with >30-point gains over strong baselines on avg@8/greedy metrics.

**Weaknesses:**

1. Evidence granularity & ablation transparency. The main text repeatedly claims surpassing original datasets and >30-point gains, but the snippet provides no per-task breakdowns, significance tests, or variance across seeds, which are crucial given exploratory randomness.
2. Top-down controls may implicitly bias discovery. While “lightweight,” Requirement Confirmation/Concept Pools could bias to familiar concepts, potentially missing novel behaviors.
3. Operational cost and failure modes of exploration. Prior work notes drift/inefficiency in bottom-up pipelines; CuES introduces memory and guidance, but we need exploration cost curves, yield vs. steps, and failure analyses (invalid/partial trajectories) to ensure practicality.

**Questions:**

1. How robust is CuES to requirement mis-specification? If Requirement Confirmation is noisy or misleading, how does performance degrade?
2. Cross-domain transfer. Can a concept pool learned in one domain help another (e.g., from AppWorld to WebShop)? I would like to see more domain transfer experiments.

---

> ### Author Response · Authors · 2025-11-16
> **Thanks to reviewer Y3MX for your careful review and time.**
>
> **Thank you very much for your time and careful review.** We address your concerns point by point below.
>
> 1. Evidence granularity & ablation transparency.
>
> We agree that analyzing the impact of exploratory randomness is important. All main results reported are averaged over multiple random seeds during the Agentic RL. And maybe your concern specifically refers to randomness originating from the data synthesis process. To quantify this, we additional conducted three independent synthesis runs under the same configuration, followed by separate training on Qwen2.5-14B-Instruct.
>
> The resulting avg@8 were **43.21%, 46.67%, and 45.14% (mean = 45.01, std = 1.73)**, all exceeding nearly a 30-point improvement over the baseline. These results demonstrate that CuES consistently produces high-quality and effective synthetic data despite the inherent stochasticity of exploration.
>
> 2. Top-down controls.
>
> We emphasize that the Top-down components in CuES, Requirement Confirmation (RC) and Concept Pools (CP), are **optional and soft-guidance mechanisms** rather than fixed constraints. They are designed to narrow irrelevant search space in realistic settings where user requirements or benchmark tasks focus on specific tool functionalities, rather than to prescribe exact tasks. The results for both Appworld and BFCL reported in the paper were not RC/CP enabled because we were concerned that the introduction of prior information would affect fairness. When disabled, CuES degenerates into a purely curiosity-driven mode and continues to autonomously discover executable goals. In our early experiments, enabling RC/CP improves exploration efficiency and benchmark-aligned coverage: under the same rollout budget, performance on avg@8 increases by 5–7 points. Therefore, RC/CP is an optional and useful design.
>
> 3. Operational cost.
>
> This is a very valuable question. We provide a quantitative study of exploration efficiency and a qualitative summary of observed failure modes. The following table reports results across three independent CuES runs also on (rollout = 50, max_turn = 30, batchsize = 30). On average, CuES produces **136 abstracted tasks from 1.5k environment steps** with a 53.7% success rate and 2.7× expansion, yielding one final usable task every 7.6 steps.
> | Run | Abs. | Succ. | Succ.% | Rew. | Exp.× | Step/Succ. | Step/Final |
> |:---:|:----:|:----:|:------:|:----:|:----:|:----------:|:----------:|
> | #1 | 125 | 69 | 55.2 | 182 | 2.64 | 21.7 | 8.2 |
> | #2 | 150 | 78 | 52.0 | 212 | 2.72 | 19.2 | 7.1 |
> | #3 | 132 | 71 | 53.8 | 204 | 2.87 | 21.1 | 7.4 |
> | avg.±std. | 135.7±12.8 | – | 53.7±1.6 | – | 2.74±0.12 | 20.7±1.3 | 7.6±0.6 |
>
>
> We also provide failure analyses. Most exploration failures fall into a few typical patterns: (1) incomplete or semantically incorrect outputs; (2) missing critical steps; (3) redundant loops; and (4) misunderstanding API scope. Most of these failure modes are due to the agent's lack of understanding of the environment. This is precisely the purpose of the agentic RL stage.
>
>
> 4. Robust.
>
> CuES is designed so that noisy or imperfect requirements are gradually normalized before they influence downstream synthesis. In RC, the user’s vague description is first processed by LLM, which rewrites and interprets the intent into a more canonical, tool-aware form. The CP then further grounds this interpreted intent into a set of environment-specific concepts (tools, entities, action motifs), effectively snapping underspecified or slightly off-target requirements back to the nearest plausible region of the task space.
>
> By the time a requirement enters the exploration, it has already been filtered and corrected once at the semantic level. During exploration and task abstraction, Environment Memory and Task Memory jointly suppress degenerate patterns through avoiding revisiting uninformative frontiers. **This combination of semantic normalization (RC + CP) and structural pruning (memories + quality control) keeps the system from getting stuck in invalid exploration modes even when the initial requirement is noisy.**
>
> 5. Cross-domain transfer.
>
> 	To evaluate Cross-domain transfer, we addtional conducted cross-domain experiments between AppWorld and BFCL. Results are summarized below.
>
> | Training Domain | AppWorld |  BFCL |
> |:---------------:|:----------------:|:-------------:|
> | **Baseline** | 11.76% | 25.74% |
> | **AppWorld** | **45.54%** | **36.43%** |
> | **BFCL** | **20.18%** | **43.00%** |
>
> These results show that **knowleage learned from one domain consistently transfer to a different domain**, even though AppWorld and BFCL differ substantially in available tools.
>
> **Your review is very valuable to our work. All of these analyses will be added to the paper or appendix to increase its practicality.** Once again, thank you for your time and careful review. If these clarifications resolve your concerns, we would be grateful if you could consider raising your score.

---

> ### Author Response · Authors · 2025-11-24
> **With about a week left in the discussion period, we remain ready to clarify any questions or concerns.**
>
> Thank you again for the detailed review. We have addressed your three main concerns: (i) for evidence granularity under synthesis randomness, we ran three independent CuES synthesis runs with identical budgets, trained three separate agents, and observed consistently strong avg@8 gains with low variance, **demonstrating stability beyond a lucky seed;** (ii) regarding possible top-down bias, we clarified that RC/CP are optional soft guidance, and were disabled in our main AppWorld/BFCL results for fairness, **CuES remains purely bottom-up and discovery-driven in these settings;** (iii) for operational cost and failure modes, we **added a quantitative efficiency table** (cost per kept task, attempts-per-accept proxy, and expansion ratio) and summarized typical failure patterns observed during exploration. We also added robustness reasoning for noisy requirements and new cross-domain transfer results between AppWorld and BFCL.
>
> **We note that the author–reviewer discussion period has about one week remaining.** If these clarifications resolve your concerns (or if any points still need further evidence), we would greatly appreciate your brief follow-up so we can address remaining issues before the discussion window closes. If these clarifications resolve your concerns, we would be grateful if you could consider raising your score.

---

> ### Author Response · Authors · 2025-11-27
> **With fewer than 5 days remaining in the author–reviewer discussion period, we wanted to follow up and ensure that all your concerns have been fully addressed.**
>
> Thank you again for your careful review and the thoughtful questions. With fewer than **5 days** remaining in the author–reviewer discussion period, we wanted to follow up and ensure that all your concerns have been fully addressed.
>
> If any part still feels unclear, especially around requirement mis-specification robustness or the mechanisms behind cross-domain transfer, we would be very grateful if you could point out what is missing, and we will clarify immediately. If our rebuttal sufficiently resolves your concerns, we respectfully ask whether you would consider **updating your score** accordingly.

---

### Author Response · Authors · 2025-11-19
**Thanks to All Reviewers — We Stand Ready to Clarify Any Doubts or Concerns**

Dear Reviewers,

We would like to sincerely **thank all of you for the time and effort** you have invested in carefully reviewing our submission and engaging in the discussion. We find your comments and suggestions extremely valuable for improving both the technical content and the presentation of our work. For example, Reviewer Y3MX’s comments on **synthesis randomness analysis**, Reviewer gNE7’s insights regarding **potential overfitting**, and Reviewer GfAy’s suggestions on **budgets and synthesis cost** have directly inspired us to refine our experiments, clarify our assumptions, and improve the organization of the paper.

As the ICLR Author–Reviewer Discussion period will end in less than two weeks, we genuinely hope that all of your concerns can be properly addressed before the deadline. If there are any remaining questions, unclear points, or additional experiments or ablations that you feel would further strengthen the paper, we would be very grateful to hear them and will do our best to respond promptly.

Thank you again for your constructive feedback and for helping us improve this work.

---

### Meta-Review · Area_Chair_eGw3 · 2025-12-30

**Summary:**

The major issue of this paper lies in the stability of the performance gain. It appears that the proposed method induces variant performance gains as the bottom-up exploration is stochastic. In addition, the robustness against mis-specification is unclear. The potential trade-off between PR and ED on WebShop also requires further explanation. Finally, all the reviewers gave negative scores with a lot of issues remaining unsolved. Thus, I suggest rejecting this paper.

**Reviewer Concerns:**

Reviewer Y3MX is concerned that CuES’s gains may be unstable due to synthesis randomness. The paper requires clearer evidence on failure modes. In addition, reviewer gNE7 has concerns about the experimental details and setups. Furthermore, reviewer GfAy’s main concern is about the efficiency analyses and more ablations.

**Reviewer Scores:**

Since most of the concerns are not well addressed yet and the initial average score is overall negative, I believe all the reviewers would possibly remain at the same score. So the overall score will still be 4, 4, 4.

---

### Decision · Program_Chairs · 2026-01-26

Reject